# Impact of MRI, CT, and Clinical Characteristics on Microbial Pathogen Detection Using CT-Guided Biopsy for Suspected Spondylodiscitis

**DOI:** 10.3390/jcm9010032

**Published:** 2019-12-21

**Authors:** Alexander Braun, Thomas Germann, Felix Wünnemann, Marc-André Weber, Marcus Schiltenwolf, Michael Akbar, Iris Burkholder, Hans-Ulrich Kauczor, Christoph Rehnitz

**Affiliations:** 1Department of Diagnostic and Interventional Radiology, University Hospital Heidelberg, 69120 Heidelberg, Germany; thomas.germann@med.uni-heidelberg.de (T.G.); felix.wuennemann@med.uni-heidelberg.de (F.W.); hans-ulrich.kauczor@med.uni-heidelberg.de (H.-U.K.); christoph.rehnitz@med.uni-heidelberg.de (C.R.); 2Institute of Diagnostic and Interventional Radiology, Pediatric Radiology and Neuroradiology, University Medicine Rostock, 18057 Rostock, Germany; marc-andre.weber@med.uni-rostock.de; 3Department of Orthopedics and Trauma Surgery, University Hospital Heidelberg, 69118 Heidelberg, Germany; marcus.schiltenwolf@med.uni-heidelberg.de (M.S.); michael.akbar@med.uni-heidelberg.de (M.A.); 4Department of Nursing and Health, University of Applied Sciences of the Saarland, 66117 Saarbruecken, Germany; iris.burkholder@htwsaar.de

**Keywords:** spondylodiscitis, biopsy, CT, MRI, CRP, microbiology

## Abstract

Spondylodiscitis accounts for 2–7% of osteomyelitis cases and is characterized by pain, systemic inflammation, and permanent neurological deficits. We aimed to identify imaging characteristics and clinical parameters to successfully predict microbiological pathogens by computed tomography (CT)-guided biopsy in suspected spondylodiscitis cases. Forty consecutive patients (mean age 65.1 years) with suspected spondylodiscitis underwent CT-guided biopsy. CT features (non-sclerotic endplate erosions (NSEs)), magnetic resonance criteria (paravertebral/epidural abscess (PA/EA) formation), and clinical data (C-reactive protein (CRP) > 50 mg/L) were assessed for their predictive potential. NSEs were detected in 6/11 (54.5%) and 1/29(3.4%) patients with positive and negative microbiology, respectively. PA and EA, respectively, were present in 7/11(63.6%) and 3/11 patients with positive microbiology and 7/29 (24.1%) and 2/29 patients with negative microbiology. CRP > 50 was observed in 7/11 (63.6%) and in 7/29 (24.1%) patients with positive and negative microbiology, respectively. Three double combinations possessed near-perfect specificity (PA + NSE, 100%; PA + CRP > 50, 96.6%; NSE + CRP > 50, 96.6%). The top three Youden indices included combinations with NSE. Since CT/magnetic resonance (MR) imaging and CRP are routinely used to evaluate spondylodiscitis, the presented diagnostic criteria and combinations can aid decision-making for biopsy.

## 1. Introduction

Spondylodiscitis accounts for 2–7% of osteomyelitis cases and has a reported annual incidence of 1/100,000 to 1/250,000 [1]. Besides intervertebral discs and adjacent vertebral bodies, spondylodiscitis often affects the surrounding paravertebral and epidural spaces. Due to the proximity of key neurological structures, such as the spinal cord and neuroforaminal nerve roots, patients not only suffer from pain and systemic inflammation but also are at risk of suffering permanent neurological deficits [2,3] and in some cases, death [4]. Therefore, a fast induction of targeted antibiotic therapy is critical and may help prevent prolonged and complicated courses [5].

Success rates of pathogen detection by blood culture can vary depending on the patient population and can reportedly be as low as 24.4% [6]. CT-guided biopsy has been used to histopathologically prove the diagnosis of spondylodiscitis and exclude malignancy [7]. Moreover, CT-guided biopsy enables the identification of causative microbiological agents, which is important for determining the appropriate antibiotic therapy. This is especially useful in cases of atypical pathogens, such as (multi-)resistant microbes or tuberculous infection [8,9]. However, CT-guided biopsy of the spine involves risks, such as pain, bleeding, and risk of neurological impairment [10,11,12]. In addition, it often fails to provide sufficient bioptic yields for pathogen detection, with reported rates of successful microbiological cultivation ranging between 28.1% [13] and 57.1% [14]. This varied success rate depends on the study design and/or preselection of the patient collective. 

Several imaging characteristics and clinical/laboratory findings have been studied in order to improve their pathogen detection capabilities. However, currently, there are no defined criteria that are clearly associated with positive/negative pathogen detection and reported assessments have a variable diagnostic performance. For example, while Ahuja et al. suggested c-reactive protein (CRP) levels > 50 mg/L as a positive predictive factor for successful pathogen detection [6], Kasalak et al. reported that CRP levels did in fact not correlate with pathogen detection [8]. Using magnetic resonance imaging (MRI), Spira et al. identified a positive correlation between paravertebral infiltration (with high specificity in cases of simultaneously present paravertebral/epidural abscess (PA/EA)) [15]. In contrast, Kasalak et al. found no significant correlation between MRI features and microbiological yield [8]. Foreman et al. investigated multiple MR/CT variables with regard to pathogen detection, such as lytic and mixed density endplate changes. However, in that study, no CT feature was identified in order to improve pathogen detection [16]. 

We, therefore, conducted this study in order to identify imaging characteristics, clinical parameters, or a combination thereof, to differentiate between successful and futile detection of microbiological pathogens by CT-guided biopsies in patients with suspected spondylodiscitis.

## 2. Experimental Section

### 2.1. Population

This retrospective study was approved by the Ethics Committee of the medical faculty at Heidelberg University (vote S-581/2014). A search of the institution’s electronic medical record database from June 2015 to March 2018 was performed for all patients who underwent CT-guided biopsy in cases of suspected spondylodiscitis. General indications for CT-guided biopsy were based on clinical orthopedic examination, laboratory tests, and imaging findings. All patients were screened for infection foci other than the spine, and none of the included patients showed a different focus other than the spine. A suggestion for a CT-guided biopsy was made after a discussion in an interdisciplinary team including orthopedic spine surgeons and senior radiologists. Patients with primary immediate indications for open surgery (for instance patients with neurologic symptoms, highly septic patients) did not undergo a CT-guided biopsy. CT-guided biopsy setting did include patients with relative indications for surgery, who had an increased surgical risk and would, therefore, benefit from the minimally invasive procedure. From an original set of 45 patients, we excluded four patients as their biopsy procedure was terminated before acquisition of sufficient bioptic material. Reasons for premature termination included the patient experiencing falling oxygen levels and intolerable pain during procedure despite optimized analgesia. One patient requested termination without any further documented explanation. Finally, a fifth patient who successfully underwent CT-guided biopsy, but could not undergo MRI examination due to an implanted pacemaker, was also excluded. Therefore, 40 consecutive patients who demonstrated clinical and morphologic findings suggestive of spondylodiscitis and who underwent CT-guided biopsy for pathogen detection, were included for analysis. In 30 of the 40 patients, additional material for histological analysis was obtained. Written informed consent was obtained from the patient or a guardian. The mean patient age was 65.1 years (range 10–89 years). As an institutional standard, in cases involving antibiotic pre-treatment (*n* = 11), antibiotic medication was stopped at least 7 days before CT-guided biopsy to increase the probability of pathogen detection. 

### 2.2. Imaging Parameters

MR imaging: All 40 patients received a pre-interventional contrast-enhanced MRI of the spine and an additional CT for biopsy planning. MRI was performed using a 3.0 tesla system (Magnetom Verio, Siemens Healthineers, Erlangen, Germany). T1-weighted turbospin-echo sequences were obtained with a 700–800 ms repetition time (TR), 12 ms echo time (TE), 150–180° flip angle, and an echo train-length of three. T2-weighted turbo spin-echo sequences were obtained by using a 2.2–7.9/113–119 ms TR/TE, a 150–180° flip angle, and an echo train-length of 19. Short tau inversion recovery sequences were obtained with a 5.1–8.9 ms TR, a 44–45 ms TE, a 210 ms inversion time, a 150–180° flip angle, and an echo train-length of 19. Contrast-enhanced T1-weighted turbo spin-echo sequences were obtained by using a 700–1000/11–12 ms TR/TE, a 150–180° flip angle, and an echo train-length of three. Gadoterate meglumine (Dotarem; Guerbet, Roissy, France) at a dose of 0.1 mmol per kg body weight was used as the intravenous contrast agent.

CT imaging and biopsy: The vertebral level and interventional approach were defined according to the pre-interventional MRI. All patients received moderate intravenous (IV) sedation using 15 mg piritramid (Dipidolor, Hameln, Germany) with continuous monitoring of their vital signs. A CT-guided biopsy was performed using a Somatom Emotion 16-slice-scanner (Siemens Healthineers, Forchheim, Germany). A thin-slice planning CT scan (0.75 mm slice thickness) was conducted in the prone position and multiplanar reconstructions were used to nontraumatically position the biopsy needle. After aseptic preparation of the skin overlying the biopsy needle trajectory, local anesthesia was administered (5–10 mL Carbostesin 0.5%, Astra Zeneca, Wedel, Germany). A small skin incision was made, and the biopsy was performed using a 3.0 mm (11 gauge) biopsy system (Bone Marrow Biopsy Needle, Somatex, Germany or Osteo-Site Ratchet, IZI Medical Products, USA). A transpedicular approach was chosen in 20/40 (50%) patients while a posterolateral/paravertebral approach was chosen in the remaining 20/40 (50%) patients. The choice of approach was based on anatomical considerations and the dominant location of the inflammatory changes. For instance, the paravertebral approach was used in cases where paravertebral inflammation was dominant. When abscesses were biopsied, additional liquefied content was aspirated. A biopsy sample was sent for microbiological examination in 40/40 (100%) patients, and an additional sample was formalin-fixed for histological analysis in 30/40 (75%) patients. Although blood cultures were not systematically performed in the patient cohort in our institution, we additionally retrospectively assessed the proportion of patients in which blood cultures were performed and whether these provided additional information on pathogen detection.

### 2.3. Evaluation of Imaging and Laboratory Findings

Laboratory findings: We assessed CRP levels using readings obtained as close as possible, but prior, to the time of CT-guided biopsy. Based on the observations by Ahuja et al. [6] we defined a group with CRP levels > 50 and a second < 50 and assessed the pathogen detection rate in both groups. Where applicable, we further assessed the patient’s antibiotic treatment regimen, and as an in-house standard, antibiotic therapy was paused at least 7 days before CT-guided biopsy. 

Imaging in general: Qualitative image analysis was performed on a standard picture archiving and communication system (Centricity PACS, Version 4.0, GE Healthcare IT solutions, Barrington, IL). It was used to select morphological images, allow coregistration and directly compare anatomic and functional data sets. All images were evaluated in consensus by two reading radiologists with 15 and 3 years of experience in the field of musculoskeletal imaging.

MR imaging: We investigated the presence or absence of PA/EA. An abscess was defined as being iso- or hypo-intense compared to muscle tissue on T1-weighted images, having a fluid-equivalent signal intensity on T2-weighted images, and displaying rim enhancement on contrast-enhanced T1-weighted fat-saturated images [17].

CT imaging: We investigated lesions of the adjacent endplates to evaluate the stage of osteitis evaluating whether non-sclerotic endplate lesions are present, as opposed to sclerotic endplate lesions or absence of any erosions. Non-sclerotic lesions may indicate an earlier stage of osteitis while sclerotic margins indicate the presence of longer-term inflammation [16,18,19]. Non-sclerotic endplate lesions were defined as lytic defects of the endplate without sclerosis of the surrounding bone.

### 2.4. Statistics

Statistical analysis was performed using the statistical package SAS for Windows Version 9.4 (SAS Institute Inc., North Carolina). All study variables were analyzed descriptively. Continuous variables were summarized using the sample size (*n*), mean, standard deviation, median, minimum, and maximum. The frequencies and percentages for each level were reported for qualitative variables. The performance of PA/EA, CRP > 50, and non-sclerotic endplate erosions (NSE) in predicting microbial pathogen-proof was described using estimates and exact 95% confidence intervals for sensitivity, specificity, and positive and negative predictive values. In addition, the predictive capacity of several combinations of variables was assessed. As a measure for a descriptive comparison between the performance of the different variables and combinations of variables the Youden Index defined as sensitivity + specificity − 1 together with the 95% confidence intervals was used [20]. The Youden Index treats false negative and false positive errors as equally undesirable. The minimal value of the Youden Index is 0 indicating that the diagnostic test reports the same proportion of positive tests for diseased and control group. If the diagnostic test produces no false positives nor false negatives, then the Youden Index reaches the maximal value 1.

## 3. Results

### 3.1. Microbiology, Histology, and Follow-Up 

Causative pathogens were detected in 11/40 patients (27.5%), including *Escherichia coli* (*n* = 3; 7.5%), *Staphylococcus* (*n* = 6; 15%), and *Streptococcus* species. (*n* = 2; 5.0%). A causative fungal agent was not detected in any of the biopsy samples. Histological analysis was performed for *n* = 30 samples and it revealed signs of acute inflammation in 9/30 patients (30.0%), chronic inflammation in 16/30 patients (53.3%), and nonspecific results in 5/30 samples (16.7%). Malignancy was excluded in all patients by either histological analysis (*n* = 30; 75.0%) or observing the improvement of symptoms and inflammatory changes in follow-up examinations (*n* = 10; 25%). In 12/40 (30%) patients additional blood cultures were performed. In 3/12 (25%) a pathogen was detected. Two detected pathogens correlated with results from CT-guided biopsy while one additional pathogen was detected in a patient with negative microbiologic result following CT-guided biopsy. This patient received targeted antibiotics and responded. Additional 16s RNA polymerase chain reaction (PCR) analyses were not performed. Empiric antibiotics were given in all culture-negative cases, three of which required additional surgical therapy. At the time of discharge from the hospital patient records and available follow-up imaging documented an improvement.

### 3.2. Laboratory Findings

CRP levels above 50 mg/L were found in 7/11 (63.6%) patients with positive microbiology but only in 7/29 (24.1%) samples with negative microbiology (Table 1). The mean CRP for all patients was 51.2 mg/L (SD 62.9). Mean CRP for patients with positive microbiology (*n* = 11) was 84.6 mg/L (SD 73.3) and for those without was 38.4 mg/L (SD 51.9).

### 3.3. Imaging Findings

MRI: PA (Figure 1) was observed in 7/11 (63.6%) patients with positive microbiology compared to only 7/29 samples (24.1%) with negative microbiology. EA (Figure 2 and Figure 3) was present in 3/11 (27.3%) patients with positive microbiology but only in 2/29 (6.9%) samples with negative microbiology. EA was also present in all patients featuring PA. 

CT: NSE (Figure 4, sclerotic erosion for better differentiation in Figure 5) was observed in 6/11 (54.5%) patients with positive microbiology but only in 1/29 (3.4%) samples with negative microbiology. 

The investigated variables varied in their respective statistical performance for predicting positive microbiological results. Sensitivity, specificity, positive predictive value (PPV), and negative predictive value (NPV) with accompanying confidence intervals for each variable are listed in Table 1. In particular, EA and NSE displayed high specificity for the prediction of positive microbiological results.

### 3.4. Combination of Variables

Double and triple combinations of included variables were tested for their sensitivity, specificity, PPV, and NPV as shown in Table 2. EA was not included as all patients with EA also exhibited PA. Performance, with respect to sensitivity, specificity, PPV, NPV, and the Youden Index, varied across the various combination of variables. Perfect specificity and PPV were achieved for the combination of PA and NSE (i.e., (PA and NSE)). Excellent specificity was achieved when using a combination of PA and CRP > 50, or CRP > 50 and NSE (i.e., (PA and CRP > 50), (CRP > 50 and NSE)). The presence of either PA or CRP > 50 resulted in very high NPV and sensitivity (i.e., (PA or CRP > 50)). The top three Youden indices included combinations with NSE. The predictive performance of all assessed variables and their combinations, as determined by calculating the Youden index, is presented in ascending order in Table 3. 

## 4. Discussion

Endeavors to identify both clinical variables and imaging features predictive of microbiological pathogen detection in spondylodiscitis patients have been undertaken by several studies [6,8,15,16,21]. However, results were, in part conflicting and there still exists no consensus in the issue. Concrete diagnostic criteria and clear clinical decision-making pathways for performing CT-guided biopsies are not yet defined. In our study, we investigated the efficacy of individual CT and MRI features as well as laboratory findings (i.e., CRP > 50) and a combination of features to predict positive pathogen detection following CT-guided biopsy. As diagnostic work-up in spondylodiscitis patients in most institutions routinely include CT and MR imaging and CRP tests, the combination of these tests alongside the presented criteria, may ease and aid in further therapeutic decision-making processes.

Since in a clinical setting not every proposed feature may be present simultaneously, we investigated a variety of combinations. Thus, for a given patient, depending on the clinical features present, physicians can tailor their decision-making based on the individual patient’s circumstances and the probability of successful pathogen detection via CT-guided biopsy. NSE and EA exhibited very high specificities as individual variables. Both features, if present, will produce very few futile CT-guided biopsies. Thus, from a clinical perspective, if either or both features are present, a biopsy can be considered under the knowledge of a potentially higher probability of pathogen detection. This holds true even more so for the combinations of PA and CRP > 50 and CRP > 50 and NSE, both of which demonstrate excellent specificity. On the other hand, the CRP > 50 or NSE combination demonstrates a high NPV and consecutively, successful pathogen detection is very unlikely in the absence of both. The combination of PA and NSE led to perfect specificity and PPV, and therefore, any patient displaying these features simultaneously might benefit from a CT-guided biopsy. The top three Youden indices included NSE as a variable. Thus, NSE can be regarded as an important feature both as a single variable and in a combination. In contrast, if NSE is not present, i.e., if the erosions were sclerosed or absent, the probability of successful pathogen detection is low. This is in line with the pathophysiologic consideration of NSE representing a more acute, active stage of osteitis while surrounding sclerosis is associated with more chronic and less active stages [18]. Foreman et al. investigated multiple MR/CT variables with regard to pathogen detection, including lytic and mixed density endplate changes [16]. Although these patterns were more frequent in the group with positive microbiology, no CT imaging parameter was identified in order to improve the prediction of microbiologic or histologic outcomes using a multivariable regression model. The authors, therefore, stated that a dedicated CT scan does not provide any additional predictive information for the biopsy result if an MRI scan is already available. Differences in our study may be explained, in part, by differences in the study population, the assessment methods, and technical differences. However, based on our results, we regard NSE as a decisive pattern, suggestive of a positive pathogen outcome. Thus, we recommend performing a CT and thoroughly evaluating the presence of NSE or on the other hand sclerotic margins/absence or erosions.

In line with our results, Spira et al. observed that epidural infiltration and paravertebral abscesses also had high specificities of 83.3% and 90.9%, respectively [15]. In our study, only a small percentage of patients featured EA and thus, EA may not represent a robust predictor for positive microbiology despite its high specificity.

Previously published studies on the predictive value of CRP levels are controversial with Kasalak et al. dismissing CRP as a reliable predictor and Ahuja et al. suggesting CRP > 50 mg/L to be linked with positive microbiological yields. Our findings support the observation from Ahuja et al. as positive pathogen detection was significantly higher in patients with CRP levels > 50 mg/L. With our own results showing solid specificity and good NPV for CRP as a single variable, such contradictory results may be attributed to divergence in the investigated patient populations or the inclusion of cases with other causes of altered CRP levels, including previous antibiotics or the time between the last antibiotic dose and the CT-guided biopsy. In our setting and as a clinical standard in our institution, we used an interval of at least 7 days between the last antibiotic dose and CT biopsy, which may also have contributed to such differences. 

None of the investigated features or combinations therein had perfect diagnostic performance values and could, therefore, be used to completely rule out the possibility for positive microbiological results. The decision to perform a biopsy should always be made in the clinical context of each individual patient. Moreover, Figure 6 presents a simplified decision-support pathway regarding CT-guided biopsy in suspected spondylodiscitis based on our results. This figure provides different probabilities of pathogen detection depending on the investigated laboratory and imaging findings in a simplified manner highlighting our main results. However, it should not be used as a final therapy-decision tool in an individual patient.

Our study had several limitations. It has to be clearly stated that this is a retrospective study based on a relatively small patient group from a single center. This study does not intend to inform clinical decision making on an individual basis. Therefore, the presented results should not be used to finally judge about performing or not performing CT-biopsies in spondylodiscitis cases without further confirmatory validation studies. Prospective studies with larger patient cohorts performed in different centers with cross-validation of the findings are needed to confirm our findings. These prospective studies might also include a multivariate analysis to investigate the weighting and interaction of each factor and predict the likelihood of having positive microbiology. We did not calculate every possible variable combination that might be associated with positive or negative biopsy yields; thus, other combinations might also be helpful in decision making. Additionally, our patient preselection, which was performed within the university-clinical setting, may vary from other possible settings (e.g., a community setting with lower frequencies of spondylodiscitis and a possibly differing spectrum of pathogens). False-positive microbiological results due to contamination with non-pathogen germ contamination cannot be ruled out with absolute certainty; however, they seem very unlikely in a scenario where clinical presentation, laboratory findings, and imaging features are all concordant with spondylodiscitis. It must also be stated that a negative pathogen detection using CT-guided biopsy does not exclude infection and should not be used as an argument to refuse antibiotic treatment. To validate our findings, further prospective studies with a larger patient cohort are needed, especially those including rare disease subgroups, such as specific spondylodiscitis. However, our study is strengthened by the fact that all data pertaining to variables used in this study are acquired prior to, or during treatment, in almost every case. Thus, the implementation of our findings into the clinical routine could prove beneficial.

## 5. Conclusions

NSE, CRP > 50, and PA/EA were associated with positive pathogen detection. A double combination of any three had near perfect specificities. If NSE, CRP > 50, and PA/EA were absent the probability of pathogen detection was low. The highest diagnostic performances were exhibited by combinations with NSE. CT evaluation for presence of NSE and sclerosed margins of erosions or absence of erosions as counterparts is therefore very informative. If these features are confirmed in future prospective studies these might support clinicians in the decision to perform or avoid a CT-guided biopsy in specific settings based on CRP, MRI, and CT. However, we would like to stress that, at the present stage, individual therapy decisions should not be based on the presented results as they have to be confirmed in future validation studies. 

## Figures and Tables

**Figure 1 jcm-09-00032-f001:**
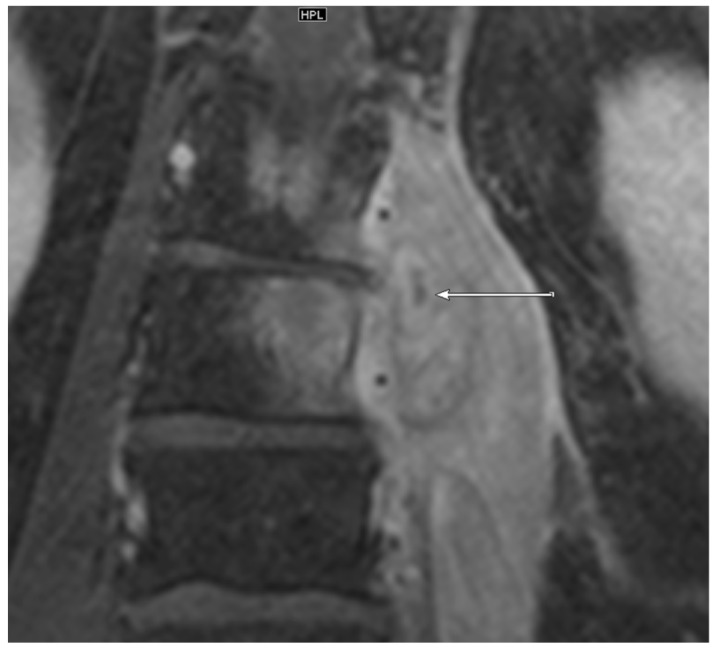
Coronal T1 weighted, fat-saturated, contrast-enhanced magnetic resonance (MR) image of the lumbar spine at the level L1-L3. A paravertebral abscess in the left psoas muscle of a 53-year old patient is present (arrow). The detected pathogen after computed CT-guided biopsy was *Staphylococcus aureus*.

**Figure 2 jcm-09-00032-f002:**
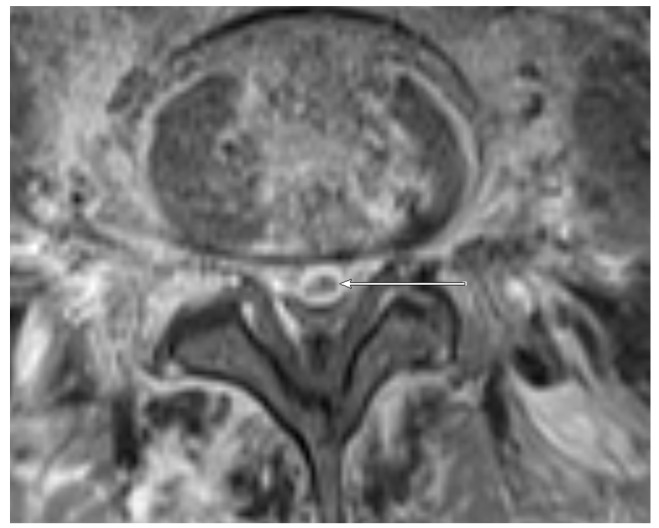
Transversal T1 weighted, fat-saturated, contrast-enhanced image of the lumbar spine level L5/L6 in a 52-year old patient. An epidural abscess is present (arrow). The detected pathogen after CT-guided biopsy was *E. coli*.

**Figure 3 jcm-09-00032-f003:**
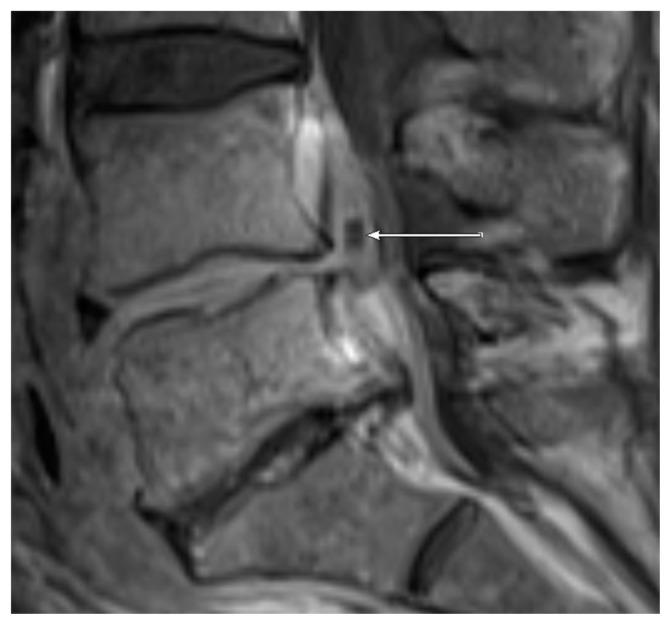
Sagittal T1 weighted, fat-saturated, contrast-enhanced image of the lumbar spine level L5-S1 in the same patient as shown in Figure 2. The epidural abscess in the spinal canal is marked with an arrow.

**Figure 4 jcm-09-00032-f004:**
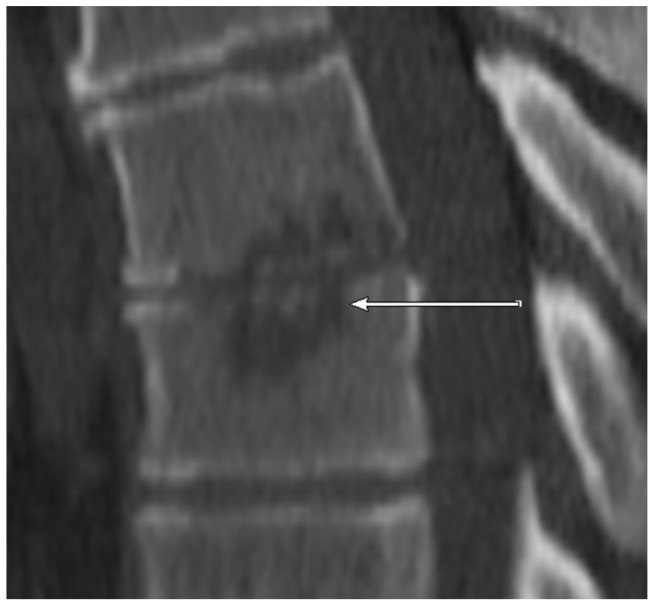
CT scan (sagittal reconstruction) of thoracic spine level Th6-Th7 showing non-sclerotic erosions of adjacent endplates of the thoracic spine in 55-year old patient. The detected pathogen after CT-guided biopsy was *E. coli*.

**Figure 5 jcm-09-00032-f005:**
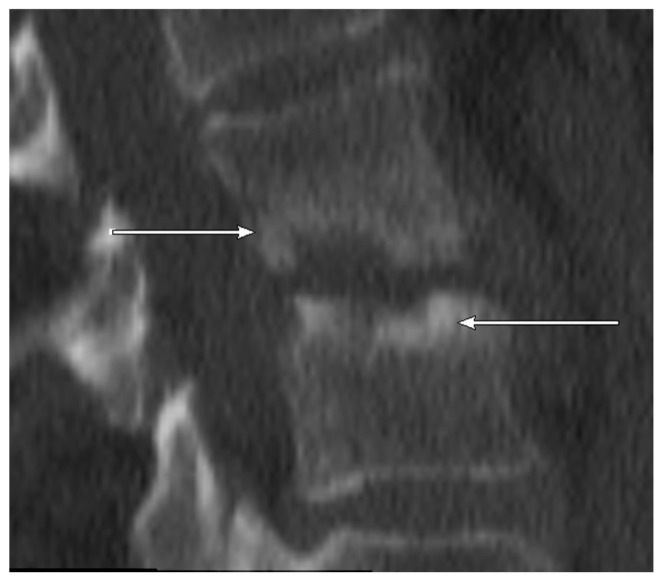
CT scan (sagittal reconstruction) of lumbar spine level L2-L3. As opposed to Figure 4 showing marginally sclerotic erosions of adjacent endplates of the lumbar spine in 77-year old patient. No detected pathogen after CT-guided biopsy.

**Figure 6 jcm-09-00032-f006:**
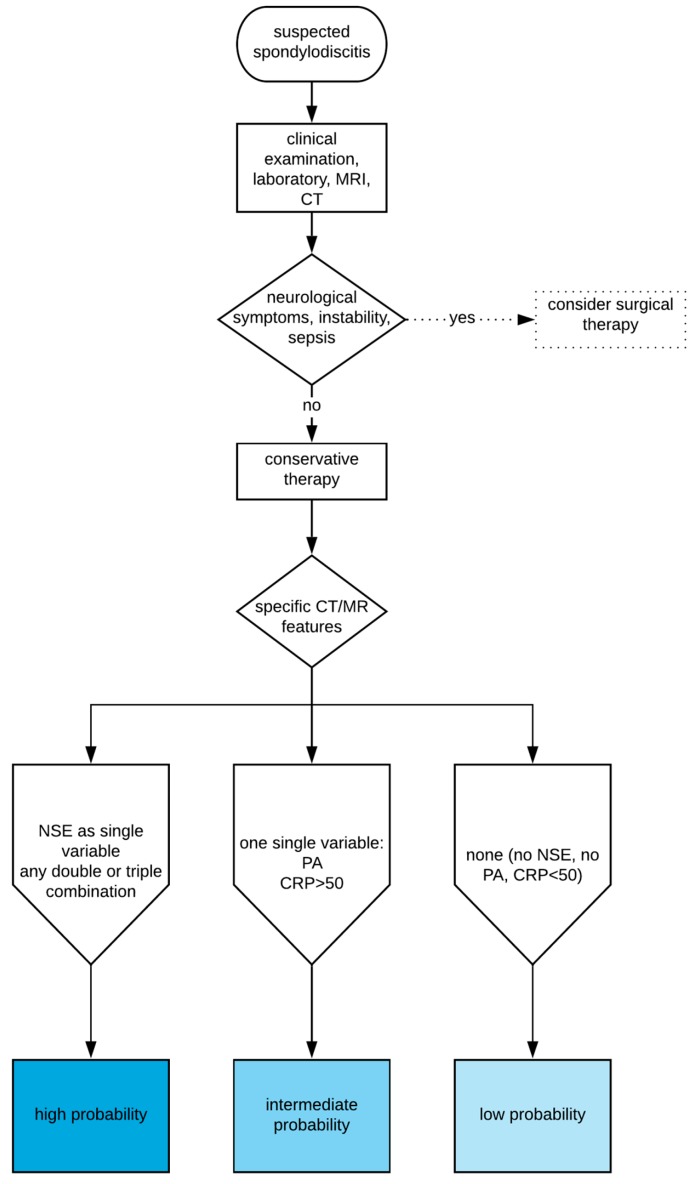
Simplified decision-support pathway regarding probability of pathogen detection following CT-guided biopsy based on imaging and laboratory findings in suspected spondylodiscitis.

**Table 1 jcm-09-00032-t001:** The performance of single variables in predicting positive microbiology from a biopsy (TP = true positive, TN = true negative, FP = false positive, FN = false negative).

Biopsy Phenotype	Sensitivity TP/(TP + FN)	Specificity TN/(TN + FP)	PPV TP/(TP + FP)	NPV TN/(TN + FN)
Paravertebral abscess	7/11 = 63.6% (30.8%–89.1%)	22/29 = 75.9% (56.5%–89.7%)	7/14 = 50.0% (23.0%–77.0%)	22/26 = 84.6% (65.1%–95.6%)
Epidural abscess	3/11 = 27.3% (6.0%–61.0%)	27/29 = 93.1% (77.2%–99.2%)	3/5 = 60.0% (14.7%–94.7%)	27/35 = 77.1% (59.9%–89.6%)
Non-sclerotic vertebral endplate erosion	6/11 = 54.6% (23.4%–83.3%)	28/29 = 96.6% (82.2%–99.9%)	6/7 = 85.7% (42.1%–99.6%)	28/33 = 84.9% (68.1%–94.9%)
CRP > 50	7/11 = 63.6% (30.8%–89.1%)	22/29 = 75.9% (56.5%–89.7%)	7/14 = 50.0% (23.0%–77.0%)	22/26 = 84.6% (65.1%–95.6%)

**Table 2 jcm-09-00032-t002:** The performance of variable combinations in predicting positive microbiology from a biopsy. (TP = true positive, TN = true negative, FP = false positive, FN = false negative)

	Sensitivity TP/(TP + FN)	Specificity TN/(TN + FP)	PPV TP/(TP + FP)	NPV TN/(TN + FN)
Paravertebral abscess and non-sclerotic erosion	4/11 = 36.4% (10.9%–69.2%)	29/29 = 100.0% (88.1%–100.0%)	4/4 = 100.0% (39.8%–100.0%)	29/36 = 80.6% (64.0%–91.8%)
Paravertebral abscess or non-sclerotic erosion	9/11 = 81.8% (48.2%–97.7%)	21/29 = 72.4% (52.8%–87.3%)	9/17 = 52.9% (27.8%–77.0%)	21/23 = 91.3% (72.0%–98.9%)
Paravertebral abscess and CRP > 50	6/11 = 54.6% (23.4%–83.3%)	28/29 = 96.6% (82.2%–99.9%)	6/7 = 85.7% (42.1%–99.6%)	28/33 = 84.8% (68.1%–94.9%)
Paravertebral abscess or CRP > 50	8/11 = 72.7% (39.0%–94.0%)	16/29 = 55.2% (35.7%–73.6%)	8/19 = 38.1% (18.1%–61.6%)	16/21 = 84.2% (60.4%–96.6%)
CRP > 50 and non-sclerotic erosion	3/11 = 27.3% (6.0%–61.0%)	28/29 = 96.6% (82.2%–99.9%)	3/4 = 75.0% (19.4%–99.4%)	28/36 = 77.8% (60.9%–89.9%)
CRP > 50 or non-sclerotic erosion	10/11 = 90.9% (58.7%–99.8%)	22/29 = 75.9% (56.5%–89.7%)	10/17 = 58.8% (32.9%–81.6%)	22/23 = 95.7% (78.1%–99.9%)
Paravertebral abscess and non-sclerotic erosion and CRP > 50	3/11 = 27.3% (6.0%–61.0%)	29/29 = 100.0% (88.1%–100.0%)	3/3 = 100.0% (29.2%–100.0%)	29/37 = 78.4% (61.8%–90.2%)
Paravertebral abscess or non-sclerotic erosion or CRP > 50	10/11 = 90.9% (58.7%–99.8%)	16/29 = 55.2% (35.7%–73.6%)	10/23 = 43.5% (23.2%–65.5%)	16/17 = 94.1% (71.3%–99.9%)

**Table 3 jcm-09-00032-t003:** The Youden Index calculated for combinations of variables in predicting positive microbiology from a biopsy in ascending order.

	Youden Index	95% Confidence Interval
Paravertebral abscess	0.201	(0.0–0.543)
Epidural abscess	0.204	(0.0–0.483)
CRP > 50 and non-sclerotic erosion	0.239	(0.0–0.510)
Paravertebral abscess and non-sclerotic erosion and CRP > 50	0.273	(0.010–0.536)
Paravertebral abscess or CRP > 50	0.279	(0.0–0.598)
Paravertebral abscess and non-sclerotic erosion	0.364	(0.080–0.648)
CRP > 50	0.395	(0.071–0.719)
Paravertebral abscess or non-sclerotic erosion or CRP > 50	0.461	(0.213–0.709)
Non-sclerotic vertebral endplate erosion	0.512	(0.210–0.814)
Paravertebral abscess and CRP > 50	0.512	(0.210–0.814)
Paravertebral abscess or non-sclerotic erosion	0.542	(0.262–0.822)
CRP > 50 or non-sclerotic erosion	0.668	(0.437–0.899)

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
