# Peer review of "Impact of MRI, CT, and Clinical Characteristics on Microbial Pathogen Detection Using CT-Guided Biopsy for Suspected Spondylodiscitis"

_jcm, 2019, doi:10.3390/jcm9010032_

Round 1

Reviewer 1 Report

This is a nice manuscript which attempts to clarify the difficult clinical decision about whether to perform CT-guided biopsy in patients with suspected SD. They attempt to stratify patients based on readily available clinical information (imaging and CRP). Their findings contradict other studies of this question. I have several comments:

Why were these three features specifically selected? Were other features first explored and then dismissed? There is no mention of blood cultures, which are helpful in the diagnosis of SD/VOM (e.g. Widdrington et al, Medical Sciences, 2018). What proportion of the cohort had bacteraemia and what proportion had blood cultures performed? What was the rationale for the CRP cut off? This seems arbitrary Whilst the authors equate culture negativity with no pathogen infection - studies of 16sPCR show that culture negative cases may still be due to infection, and around 25% of SD/VOM cases are culture negative yet respond to antibiotic treatment. The authors need to provide follow up information on what happened to the culture negative cases in their cohort - did they progress, were they treated anyway for culture-negative SD/VOM with antibiotics? Did they do 16sRNA PCR to detect occult pathogens? Much more information on the Youden index is needed (reference, validation, how it was calculated, what criteria were used etc) is needed (see PMID: 16161804) The actual data need to be shown in Table 2, not just the sensitivity/specificity etc. It is important to see how many culture positive cases would be missed by these combinations.   What was the predictive value of PA or NSE and CRP>50? There needs to be a multivariate analysis (these factors probably interact) I am troubled by the suggestion from the authors that this combination of factors could be used clinically as a tool to decide on whether to biopsy or not (see Fig. 5). SD/VOM is notoriously difficult to diagnose, and our default position in the absence of a positive blood culture is to perform a biopsy. To conclude that biopsy can be avoided in a proportion of people, based on a very small sample of retrospective data from a single centre without any cross validation, and in the face of existing contradictory data, is not scientifically justifiable. The authors need to be more cautious about their interpretation of these data and make absolutely clear that this set of factors should not be used to inform clinical decision making without further, more statistically robust, prospective validation studies. 

Author Response

General response to the queries:

We greatly appreciate the effort of the editor and the reviewers in reviewing this manuscript and in providing constructive recommendations for its improvement. We have adhered to all their recommendations (please see our responses to each query) as much as possible and have stated our position on each of the suggestions, item by item, in the following statement of improvement. All changes in the revised manuscript are visible in the “track changes” mode in MSWord. The changes are also detailed and highlighted in the statement of improvement (bold and underlined) and in the manuscript file.

Reviewer #1:

Reviewer #1, comment 1

This is a nice manuscript which attempts to clarify the difficult clinical decision about whether to perform CT-guided biopsy in patients with suspected SD. They attempt to stratify patients based on readily available clinical information (imaging and CRP). Their findings contradict other studies of this question. I have several comments:

Author response comment 1

We thank the reviewer for the kind opening statement and hope that our adjustments to the manuscript help to reach the anticipated degree of finalization.

Author action comment 1

No action needed.

Reviewer #1, comment 2

Why were these three features specifically selected? Were other features first explored and then dismissed?

Author response comment 2

We limited the number of features investigated in order to keep this study simple and comprehensive. From a clinical perspective the presence of abscesses and the CRP levels play an important and central role in therapy decision making in inflammatory diseases in general and in particular in cases of spondylodiscitis. Therefore these two items reflect every-day clinical questions. Both have also been discussed in previous studies in other settings. On the other hand, non-sclerotic erosions – as seen on CT-imaging- have not gained much emphasis in the literature so far. However, in the routine practice, we became aware that there might be differences between patients displaying NSE compared to whose with sclerotic margins, which was a central question of this study. Our intention when conducting the study was also to not expand imaging criteria to features that have been tested previously and had no influence (e.g. disc height etc.) One main selection criterion was only to include data which is readily available in clinical routine (lab, CT, MRI). There were no other features explored or dismissed.

Author action comment 2

No specific action needed.

Reviewer #1, comment 3

There is no mention of blood cultures, which are helpful in the diagnosis of SD/VOM (e.g. Widdrington et al, Medical Sciences, 2018). What proportion of the cohort had bacteraemia and what proportion had blood cultures performed? 

Author response comment 3

We fully agree that blood cultures are also helpful in the diagnosis of SD/VOM. Out of 41 subjects included in our study, blood culture results were available for 12 patients. Out of those 12, bacterial growth could be detected in 3 samples (25%). In 2/3 patients the positive blood culture revealed the same pathogen as the CT-biopsy and therefore correlated with CT-biopsy results. In 1 patient blood culture led to an additional pathogen detection in a patient with negative microbiology following CT-biopsy. In our opinion this underlines the potential additional role of the blood culture in the setting of SD. In concordance with the mentioned literature the blood culture is only positive in a part of patients but is easy to perform and may be used as an additional tool and not as a replacement of the CT-guided biopsy. (

Author action comment 3

To fully adhere to the reviewer`s comment we have made the following changes in the manuscript:

In the Experimental Section on page 3 we added:

“Although blood cultures were not systematically performed in the patient cohort in our institution, we additionally retrospectively assessed the proportion of patients in which blood cultures were performed and whether these provided additional information on pathogen detection.”

In the Results on page 4 we added:

In 12/40 (30%) patients additional blood cultures were performed. In 3/12 (25%) a pathogen was detected. 2 detected pathogens correlated with results from CT-guided biopsy while 1 additional pathogen was detected in a patient with negative microbiologic result following CT-guided biopsy“. 

Reviewer #1, comment 4

What was the rationale for the CRP cut off? This seems arbitrary whilst the authors equate culture negativity with no pathogen infection - studies of 16sPCR show that culture negative cases may still be due to infection, and around 25% of SD/VOM cases are culture negative yet respond to antibiotic treatment.

Author response comment 4

CRP cut off as a possible predictor has been discussed controversially (with some authors finding no correlation at all). The reason for investigating CRP>50 was a recent paper by Ahuja et al. (The effectiveness of computed tomography- guided biopsy for the diagnosis of spondylodiscitis: an analysis of variables affecting the outcome; European Review for Medical and Pharmacological Sciences; 2017; 21: 2021-2026) who found CRP>50mg/dl to be associated with successful pathogen detection. Our observation therefore supports the results of Ahuja et al. Furthermore, we fully agree with the reviewer that culture negativity does not mean that there is no infection. The response to antibiotics is one of the observations that speak for that.   It was not our intent to suggest synonymity between futile pathogene detection and no infection. We only referred to the results of the pathogen detection using CT-guided biopsy. It should clearly be stated that a negative pathogen detection using CT-guided biopsy does not exclude an infection and should not be used as an argument to refuse antibiotic treatment.

Author action comment 4

In the Experimental section on page 3 we added:

“Based on the observations by Ahuja et al. [6] we defined a group with CRP levels > 50 and a second <50 and assessed the pathogen detection rate in both groups.”

In the Discussion on page 11 we added:

“Our findings support the observation from Ahuja et al. as positive pathogen detection was significantly higher in patients with CRP levels >50 mg/l.”

In the limitation paragraph we added the following statement:

“It must also be stated that a negative pathogen detection using CT-guided biopsy does not exclude an infection and should not be used as an argument to refuse antibiotic treatment.”

Reviewer #1, comment 5

The authors need to provide follow up information on what happened to the culture negative cases in their cohort - did they progress, were they treated anyway for culture-negative SD/VOM with antibiotics? Did they do 16sRNA PCR to detect occult pathogens?

Author response comment 5

As also stated in response to comment 4 blood cultures were additionally performed in 12 patients. This lead to an additional pathogen detection in 1 case with subsequent antibiotic treatment and clinical response. All other culture negative cases were treated with empirical antibiotic therapy, three of which required additional surgical therapy. At the time of discharge from the hospital patient records and available follow-up imaging documented an improvement.    As correctly mentioned, 16s RNA PCR is a possibility to detect occult pathogens especially in patients with negative microbiology. However, unlike bacterial cultivation, it lacks the possibility for simultaneous resistance testing. (e.g. Sheikh et al. Front. Cell. Infect. Microbiol. 7:60). 16s RNA PCR is yet not implemented in clinical routine and has not been performed in any of the patients in our cohort

Author action comment 5

We added these pieces of information in the results section as follows:

The section caption on page 4 now reads:

3.1. Microbiology, histology and follow-up

The respective paragraph together with the paragraph on blood culture on page 4 now reads:

In 12/40 (30%) patients additional blood cultures were performed. In 3/12 (25%) a pathogen was detected. Two detected pathogens correlated with results from CT-guided biopsy while 1 additional pathogen was detected in a patient with negative microbiologic result following CT-guided biopsy. This patient received targeted antibiotics and responded. Additional 16s RNA polymerase chain reaction (PCR) analyses were not performed. Empiric antibiotics were given in all culture negative cases, three of which required additional surgical therapy. At the time of discharge from the hospital patient records and available follow-up imaging documented an improvement.”    

Reviewer #1, comment 6

Much more information on the Youden index is needed (reference, validation, how it was calculated, what criteria were used etc) is needed (see PMID: 16161804).

Author response comment 6

We agree and added a more detailed explanation of the Youden Index at the end of chapter 2.4 and the corresponding reference.

Author action comment 6

In the Statistics section on page 4 we added:

“As measure for a descriptive comparison between the performance of the different variables and combinations of variables theThe Youden Index defined as sensitivity+specificity-1 and together with the 95% confidence intervals were calculated for each variable and each combination of variables were used [20]. The Youden Index treats false negative and false positive errors as equally undesirable. The minimal value of the Youden Index is 0 indicating that the diagnostic test reports the same proportion of positive tests for diseased and control group. If the diagnostic test produces no false positives nor false negatives then the Youden Index reaches the maximal value 1.”

Reviewer #1, comment 7

The actual data need to be shown in Table 2, not just the sensitivity/specificity etc. It is important to see how many culture positive cases would be missed by these combinations.

Author response comment 7

We agree and revised the table 2 accordingly as requested. For consistency reasons we now also provide the table 1 with the actual data.

Author action comment 7

Table 2 was changed accordingly to this comment and actual data were added in all cells of the table. Likewise Table 1 was completed for consistency reasons. Please see our changes in the tables in the revised manuscript.

Besides, to increase consistency the cells have been highlighted with colors that match to the colors in Figure 5 indicating different probabilities of pathogen detection.

Reviewer #1, comment 8

What was the predictive value of PA or NSE and CRP>50? There needs to be a multivariate analysis (these factors probably interact) I am troubled by the suggestion from the authors that this combination of factors could be used clinically as a tool to decide on whether to biopsy or not (see Fig. 5).

Author response comment 8

We thank the reviewer for pointing this out. In fact the reviewers` comment correctly touches different points that need to be discussed. First this inclusion of an “and/or” combination ([PA and CRP>50] or NSE) as we did in the original manuscript might lead to confusion. We did only analyze this specific “and/or” combination and refrained from analyzing other combinations like the one mentioned by the reviewer (PA or NSE and CRP>50). We decided to omit this combination and believe that it does not impair our results. Second we agree that decision making should not be based on any of our presented combinations without further confirmatory validation studies. Please see our response and changes following the comment 9 of the reviewer 1.   Third we would like to provide a discussion on the multivariate analysis. We agree that the factors considered probably interact and that it would be interesting to predict the likelihood of having a positive microbiology by using a multivariate logistic model. In fact, recommendations for the number of events needed per variable in a multivariate logistic regression model vary between 10 (Peduzzi P, Concato J, Kemper E, Holford TR, Feinstein AR. A simulation study of the number of events per variable in logistic regression analysis. J Clin Epidemiol. 1996;49(2):1373–1379) and 20 (Austin PC, Steyerberg EW: Events per variable (EPV) and the relative performance of different strategies for estimating the out-of-sample validity of logistic regression models. Stat Methods Med Res. 2017 Apr; 26(2):796-808.). In any case the number of patients with positive microbiology in the underlying study was 11 and appeared too small to get reasonable and accurate estimates from multivariate logistic modeling. This is also reflected in the response to comment 9 of the reviewer 1: The data base in our study which included 40 patients with retrospective analysis is too small to draw final conclusions and in this aspect also to calculate a valid multivariate logistic model. In a prospective study with larger cohorts it would definitely make sense to investigate the weighting and interaction of these factors.

Author action comment 8

First we omitted the data in the Abstract and Results section concerning the ([PA and CRP>50] or NSE) combination. We deleted the respective sentences on page 9 as well as the last rows of the table 2 and table 3. In the Discussion on page 10 the following sentence was deleted:

“The highest Youden index, indicative of the best statistical overall performance, was achieved by detecting PA and CRP>50, or otherwise NSE (i.e., [PA and CRP>50] or NSE). This combination led to high sensitivity and PPV achieved both excellent specificity and NPV.”  

As discussed in comment 9 of the reviewer 1 changes were applied to the figure 5 and all statements concerning specific recommendations to perform or not perform a biopsy (please see the response and changes there and in the revised manuscript).

In the Limitation paragraph of the discussion we additionally express the usefulness to perform a multivariate analysis in order to investigate the weighting and interaction of these factors:

“These prospective studies might also include a multivariate analysis to investigate the weighting and interaction of each factor and predict the likelihood of having a positive microbiology”

Reviewer #1, comment 9

SD/VOM is notoriously difficult to diagnose, and our default position in the absence of a positive blood culture is to perform a biopsy. To conclude that biopsy can be avoided in a proportion of people, based on a very small sample of retrospective data from a single centre without any cross validation, and in the face of existing contradictory data, is not scientifically justifiable. The authors need to be more cautious about their interpretation of these data and make absolutely clear that this set of factors should not be used to inform clinical decision making without further, more statistically robust, prospective validation studies.

Author response comment 9

We thank the reviewer for this comment and fully agree that the diagnosis of SD/VOM is difficult and that we have to interpret our data more carefully. Our data are based on a small patient cohort in a single center and were analyzed retrospectively. This study did not aim to finally state on the therapy decision making in individual patients. We therefore rephrased the sentences in the Discussion which included too straight statement towards a recommendation to perform or not perform a biopsy. Also we changed the figure 5. We deleted the last row which included the suggestion if a CT-biopsy is recommended or not and revised the figure description. We now only state on the probability of pathogen detection in our cohort based on imaging/laboratory findings in a manner of a decision-support instead of a decision pathway regarding perormaing a CT-biopsy.

Following the advice of the reviewer we have also revised the limitation paragraph which now clearly states the retrospective data base and that no individual therapy decision should be based on our statements without further validation.

Author action comment 9

In the Discussion on page 10 the following sentences were revised:

Thus, from a clinical perspective, if either or both features are present, a biopsy can be considered under the knowledge of a potentially higher probability of pathogen detection.”

“The combination of PA and NSE led to perfect specificity and PPV, and therefore, any patient displaying these features simultaneously might benefit from a CT-guided biopsy.“

We deleted the following statement on page 11 of the discussion:

; however, the presented signs and combinations can be used to guide physicians in their decision-making.

The figure 5 was revised as follows:

The description of the figure 5 was revised as:

“Figure 5. Simplified decision-support pathway regarding probability of pathogen detection following CT-guided biopsy based on imaging and laboratory findings in suspected spondylodiscitis.”

In the Limitation paragraph of the Discussion we now clearly state:

“Our study had several limitations. First, it has to be clearly stated that this is a retrospective study based on a relatively small patient group from a single center. This study does not intent to inform clinical decision making on an individual basis. Therefore the presented results should not be used to finally judge about performing or not performing CT-biopsies in spondylodiscitis cases without further confirmatory validation studies. Prospective studies with larger patient cohorts performed in different centers with cross-validation of the findings are needed to confirm our findings.

We also completely revised the conclusion:

NSE, CRP>50 and PA/EA were associated with positive pathogen detection. A double combination of any three had near perfect specificities. If NSE, CRP>50 and PA/EA were absent the probability of pathogen detection was low. The highest diagnostic performances were exhibited by combinations with NSE. CT evaluation for presence of NSE and sclerosed margins of erosions or absence of erosions as counterparts is therefore very informative. If these features are confirmed in future prospective studies these might support clinicians in the decision to perform or avoid a CT-guided biopsy in specific settings based on CRP, MRI and CT. However, we would like to stress that, at the present stage, individual therapy decisions should not be based on the presented results as they have to be confirmed in future validation studies.

Reviewer 2 Report

The authors report an interesting study on the analysis of different clinical and imaging parameters for the decisionmaking process during the management of patients affected by suspected spondylodiscitis. They found that the non-sclerotic end-plate erosions, C-reactive protein >50 and paravertebral/epidural abscess were associated with positive pathogen detection. A double combination of any three had near-perfect specificities, and therefore, CT-guided biopsy was highly recommended for patients suspected to have spondylodiscitis. The topic is interesting, the paper is well written, the methodology is correct. Conclusions are supported by results. 

Author Response

Statement of improvement

Manuscript jcm-660917 titled "Impact of MRI, CT, and clinical characteristics on microbial pathogen detection using CT-guided biopsy for suspected spondylodiscitis”

General response to the queries:

We greatly appreciate the effort of the editor and the reviewers in reviewing this manuscript and in providing constructive recommendations for its improvement. We have adhered to all their recommendations (please see our responses to each query) as much as possible and have stated our position on each of the suggestions, item by item, in the following statement of improvement. All changes in the revised manuscript are visible in the “track changes” mode in MSWord. The changes are also detailed and highlighted in the statement of improvement (bold and underlined) and in the manuscript file.

Reviewer #2:

Reviewer #2, comment 1

The authors report an interesting study on the analysis of different clinical and imaging parameters for the decisionmaking process during the management of patients affected by suspected spondylodiscitis. They found that the non-sclerotic end-plate erosions, C-reactive protein >50 and paravertebral/epidural abscess were associated with positive pathogen detection. A double combination of any three had near-perfect specificities, and therefore, CT-guided biopsy was highly recommended for patients suspected to have spondylodiscitis. The topic is interesting, the paper is well written, the methodology is correct. Conclusions are supported by results.

Author response comment 1

We thank the reviewer very much for this comment and hope that the final version will be able to justify the expressed appreciation for our work.

Author action comment 1

No specific action needed.

Round 2

Reviewer 1 Report

The authors have addressed most of these comments satisfactorily.

There is a stray "Moreover" on line 327.